# Development of a novel real-time polymerase chain reaction assay for the sensitive detection of *Schistosoma japonicum* in human stool

**Sara Halili**[1,2]*, **Jessica R. Grant**[1], **Nils Pilotte**[1,3], **Catherine A. Gordon**[4], **Steven A. Williams**[1,2,5]

**1** Department of Biological Sciences, Smith College, Northampton, Massachusetts, United States of America, **2** Program in Biochemistry, Smith College, Northampton, Massachusetts, United States of America, **3** Department of Biological Sciences, Quinnipiac University, Hamden, Connecticut, United States of America, **4** QIMR Berghofer Institute of Medical Research, Molecular Parasitology Laboratory, Brisbane, Australia, **5** Molecular and Cellular Biology Program, University of Massachusetts, Amherst, Massachusetts, United States of America

* shalili@smith.edu

**Data Availability Statement:** Sequence file is available from GenBank with accession number

## Abstract

### Background

Elimination and control of *Schistosoma japonicum*, the most virulent of the schistosomiasis-causing blood flukes, requires the development of sensitive and specific diagnostic tools capable of providing an accurate measurement of the infection prevalence in endemic areas. Typically, detection of *S. japonicum* has occurred using the Kato-Katz technique, but this methodology, which requires skilled microscopists, has been shown to radically under-estimate levels of infection. With the ever-improving capabilities of next-generation sequencing and bioinformatic analysis tools, identification of satellite sequences and other highly repetitive genomic elements for use as real-time PCR diagnostic targets is becoming increasingly common. Assays developed using these targets have the ability to improve the sensitivity and specificity of results for epidemiological studies that can in turn be used to inform mass drug administration and programmatic decision making.

### Methodology/Principal findings

Utilizing Tandem Repeat Analyzer (TAREAN) and RepeatExplorer2, a cluster-based analysis of the *S. japonicum* genome was performed and a tandemly arranged genomic repeat, which we named SjTR1 (*Schistosoma japonicum* Tandem Repeat 1), was selected as the target for a real-time PCR diagnostic assay. Based on these analyses, a primer/probe set was designed and the assay was optimized. The resulting real-time PCR test was shown to reliably detect as little as 200 ag of *S. japonicum* genomic DNA and as little as 1 egg per gram of human stool. Based on these results, the index assay reported in this manuscript is more sensitive than previously published real-time PCR assays for the detection of *S. japonicum*.

MW631938. All other relevant data are within the manuscript and its Supporting Information files.

**Funding:** This work was supported by the National Academy of Sciences, USA (through the Blakeslee Fund for Genetics Research awarded to Smith College). The Philippines field work study was supported by grants from UBS Optimus foundation (ID496600), the NHMRC program grant (ID1037304), and the NHMRC project grant (ID613671). The funders had no role in study design, data collection and analysis, decision to publish, or preparation of the manuscript.

**Competing interests:** The authors have declared that no competing interests exist.

## Conclusions/Significance

The extremely sensitive and specific diagnostic assay described in this manuscript will facilitate the accurate detection of *S. japonicum*, particularly in regions with low levels of endemicity. This assay will be useful in providing data to inform programmatic decision makers, aiding disease control and elimination efforts.

### Author summary

Schistosomiasis is a Neglected Tropical Disease (NTD) estimated to infect more than 230 million people worldwide. Of the various species of schistosomes that cause disease in humans, *Schistosoma japonicum* is considered the most virulent. As such, this pathogen presents a crucial public health threat. Typically, diagnosis of *S. japonicum* has been performed via the Kato-Katz technique which has been shown to dramatically underestimate the burden of infection, resulting in a need for improved detection strategies. To address this need, we have employed bioinformatic tools in order to identify a tandemly arranged, highly repetitive, DNA sequence, SjTR1, in the *S. japonicum* genome. Utilizing this sequence as a real-time PCR assay target, we have developed a sensitive and specific assay for the detection of *S. japonicum* DNA. Employment of this assay in field settings will facilitate the accurate detection of *S. japonicum* and provide guidance capable of informing mass drug administration efforts targeting elimination.

## Introduction

Schistosomiasis is a debilitating Neglected Tropical Disease (NTD) estimated to infect more than 230 million people worldwide. Of the blood flukes causing this disease, *Schistosoma japonicum*, which is endemic in China, Indonesia, Taiwan (zoophilic strain), and the Philippines, is the most virulent species [1–3]. *S. japonicum* is known to successfully infect 46 mammalian hosts [4] with each adult female schistosome producing approximately 1,000 eggs per day [5]. The deposition of *S. japonicum* eggs can result in the formation of tissue granulomas, in turn giving rise to a severe immune response that frequently leads to cognitive impairment, growth stunting, diarrhea, rectal bleeding and abdominal pain, with some people developing severe hepatosplenic disease [1,6].

While *S. japonicum* infection in both children and adults can be controlled through periodic administration of the preventive chemotherapeutic praziquantel [7], until now *S. japonicum* has only been successfully eliminated from Japan [8]. *S. japonicum's* wide range of animal hosts make its elimination highly challenging [9]. This pathogen's complicated life cycle, involving the excretion of eggs into freshwater environments followed by further growth in *Oncomelania hupensis*, a freshwater snail that serves as an intermediate host [9], results in the exposure of many reservoir hosts—such as bovines, rodents, goats, and dogs—to infection [10–11]. Given these challenges, elimination will require accurate, efficient, and sensitive diagnostic tools that can be applied to screening humans, other definitive hosts, and snail vectors.

Typically, detection of *S. japonicum* eggs from human-derived stool samples has occurred using the quantitative, coprological Kato-Katz technique [12–13]. However, this technique, which requires skilled microscopists, has been shown to have low reproducibility in determining egg counts [14] and to have poor sensitivity in low endemicity/infection intensity settings [15]. In contrast, molecular techniques such as real-time PCR can detect extremely limited

**Table 1.  *S. japonicum* assay primers and probe.**

| | |
|---|---|
| Forward Primer | 5'-TGT CGT GCA CAA CCT TCT TC-3' |
| Reverse Primer | 5'-ACA ACT CAT CAC CGC CAA TC-3' |
| Probe | 5'-/56-FAM/ TGG CGA GAT / ZEN/ GTT GTG GGT GTA AGT / 3IABkFQ/-3' |

concentrations of parasite-derived DNA, exhibiting greater sensitivity than microscopy-based approaches [16]. While a small number of *S. japonicum*-targeting real-time PCR diagnostic assays can be found in the literature, each assay has targeted comparatively low copy number sequences such as the mitochondrial NADH dehydrogenase I gene [17], a putative DNA photo-lyase gene [18], and the mitochondrial DNA 16S rRNA gene [19]. While these assays have exhibited greater sensitivity than microscopy-based techniques, targeting coding sequences inevitably raises concerns about target specificity as such genes frequently exhibit greater conservation among closely related species [20]. In contrast, non-coding tandemly repeated genomic regions make ideal real-time PCR targets, often providing high sensitivity and species-level specificity [20–21]. Here we report the development of a highly sensitive, specific real-time PCR assay for the detection of *S. japonicum* in human stool by targeting an abundant, tandemly-repeated genomic DNA sequence, which we have named SjTR1 (*Schistosoma japonicum* Tandem Repeat 1) (GenBank:MW631938).

## Materials and methods

### Ethics statement

Informed written consent was received from all human participants in the study and ethical approval was provided by the Ethics Committee of the Research Institute of Tropical Medicine (RITM), Manila, and the Queensland Institute of Medical Research (QIMR) Human Research Ethics Committee (Approval Number: H0309-058 (P524)).

### Assay target identification

A library of raw, paired-end reads resulting from the next-generation sequencing of female *S. japonicum*-derived DNA (SRR6841388) was retrieved from the Sequence Read Archive at the National Center for Biotechnology Information (NCBI) website (www.ncbi.nlm.nih.gov). Following quality analysis using FASTQC, a random sampling of 500,000 sequences was analyzed using Tandem Repeat Analyzer (TAREAN) and RepeatExplorer2 software as implemented on the Galaxy-based web server [22–26]. The repetitive elements of the genome were characterized using a graph-based technique, which forms clusters of reads that demonstrate 90% or greater identity over 55% or greater of the longer sequence length. Clusters containing threshold-exceeding numbers of shared paired-end reads were then aligned to form superclusters, resulting in the identification of multiple, high-coverage consensus sequences [24–26]. From these genomic elements, the most repetitive sequence (SjTR1) was chosen as a putative real-time PCR assay target.

### Real-time PCR assay design

A set of forward and reverse primers and a probe (Table 1) were designed using PrimerQuest Tool software (Integrated DNA Technologies, Coralville, IA) to target the SjTR1 sequence (Fig 1). The primers and probes were analyzed with the Primer-Blast tool, available from the NCBI website (https://www.ncbi.nlm.nih.gov/tools/primer-blast/) to screen for possible off-target amplification sites. The probe was labeled with a 56-FAM fluorophore at the 5' end and was double quenched with ZEN and 3IABkFQ.

SjTR1 Sequence

5'-TAGGGGGGAATGTCGTGCACAACCTTCTTCCCCATATAAAGATGATGGCGAGATGTTGTGGGTGTAAGTTGATTGGCGGTGATGAGTTGTGCATGAAAAAA-3'

Forward Primer Probe Reverse Primer

**Fig 1. The amplified region of the SjTR1 genomic sequence.** Locations of the *S. japonicum* assay primer and probe binding sites are indicated.

## Nucleic acid extraction and whole genome amplification of *S. japonicum, S. mansoni, S. haematobium,* and *S. mekongi* genomic DNA

DNA was extracted from samples of *S. japonicum* (Hubei, China), *S. mansoni* (Senegal River Basin), *and S. haematobium* (Zanzibar Island, Tanzania) single male worms graciously provided by the Natural History Museum, London, UK [27]. These extractions were performed using the Isolate II Genomic DNA kit from Meridian Bioscience (Memphis, TN) and extracted DNA was whole genome amplified using the REPLI-g UltraFast Mini Kit (Qiagen, Germantown, MD) in accordance with manufacturer protocols. DNA from *S. mekongi* worms (Maintained at Applied Malacology Laboratory, Faculty of Tropical Medicine, Mahidol University, Thailand) was extracted using the DNeasy Blood & Tissue Kit (Qiagen, Germantown, MD).

## Real-time PCR assay optimization and limit of detection

Real-time PCR assay optimization experiments were performed as described in previous studies [26] using the StepOnePlus Real-Time PCR System (Thermofisher Scientific, Waltham, MA). Briefly, the optimal annealing/extension temperature for the assay was determined by testing a range of annealing temperatures from 52˚C to 62˚C. We initially tested six temperatures at 2˚C increments from 52˚C to 62˚C and then narrowed the range, testing six temperatures at 0.5˚C increments from 56.5˚C to 59˚C. The annealing temperature yielding the lowest Cq value was chosen. Optimal primer concentrations were determined using 7 μL reactions containing 3.5 μL of TaqPath ProAmp Master Mix (Thermofisher Scientific), and employing doubling dilutions of forward and reverse primers at concentrations ranging from 62.5 nM to 1000 nM in all possible combinations. The limit of detection was determined under optimized assay conditions by testing ten-fold serial dilutions of a *S. japonicum* genomic DNA stock with template masses ranging from 100 pg to 1 ag. Negative controls for all experiments were prepared using the same master mix without any template DNA added.

## Specificity panel for the *S. japonicum* assay

Specificity of the assay was assessed by testing against a variety of non-target DNA templates. All testing occurred in 7 μL reactions containing 3.5 μL of TaqPath ProAmp Master Mix and optimized primer/probe concentrations. All templates were added at a mass of 200 pg/reaction and included isolated DNA from *S. mansoni, S. haematobium, S. mekongi, Trichuris trichiura, Strongyloides stercoralis, Ancylostoma duodenale, Ascaris lumbricoides, Baylisascaris procyonis, Ancylostoma caninum, Parascaris univalens, Anisakis typica, Necator americanus, Taenia solium, Taenia crassiceps, Giardia intestinalis,* Mock Microbial Community B(HM-277D) (BEI resources, Manassas,VA), *Candida albicans (strain L26), Escherechia coli,* and human DNA. Origins of samples are provided in S1 Table.

## Generation of plasmid control containing the assay target sequence

SjTR1 qPCR forward and reverse primers were used to amplify the target sequence from genomic *S. japonicum* DNA using conventional PCR. The target sequence was size selected using agarose

gel electrophoresis, purified from the gel using the Monarch Genomic DNA Purification Kit (New England Biolabs, Ipswich, MA), and cloned using the Zero Blunt TOPO PCR Cloning Kit (Invitrogen, Carlsbad, CA) as previously described [26]. The resulting plasmid was used to transform NEB Express Competent *E. coli* cells (New England Biolabs), which were plated on LB-kanamycin plates and grown overnight at 37˚C. Following the selection of colonies, PCR and sequencing were performed to identify a plasmid containing a single copy of the assay's target sequence which was subsequently used as a positive control and to calculate the assay's efficiency.

### Real-time PCR assay efficiency

To determine assay efficiency, the plasmid concentration was quantified using a Qubit 2.0 Fluorometer (Thermofisher Scientific), and serial dilutions were created resulting in stocks spanning 10-fold serial dilutions ranging from 100 pg/μL to 100 ag/μL. The number of plasmid copies was calculated for each dilution with 100 ag of *S. japonicum* DNA corresponding to 26 copies of the target sequence. The log of plasmid copies was plotted against the mean Cq resulting from the amplification of each concentration of plasmid (11 replicates per concentration) and the slope of the linear line was used to calculate the assay's efficiency.

### Testing naive human stool spiked with *S. japonicum* eggs

A series of Lysing Matrix E tubes (MP Biomedicals, Santa Ana, CA) were prepared according to the manufacturer's recommendations. Three samples each of 1, 3 and 10 *S. japonicum* eggs were transferred to their respective Lysing Matrix E tubes. Using a sterile loop, 50 mg of naive stool (BioIVT, Westbury, NY) was then added to each tube, resulting in three samples containing 20, 60, and 200 eggs per gram (epg), respectively. Three samples were prepared for each concentration of eggs. Following preparation, all samples were homogenized for 40 seconds using the FastPrep-24 5G Instrument (MP Biomedicals) on a speed setting of 6, and DNA extraction was performed as described previously using the FastDNA Spin Kit for Soil (MP Biomedicals) [26]. To test the assay's ability to detect as little as 1 epg of stool, additional replicates were prepared by spiking a single egg into each of ten 1 g stool samples. The weight of the stool was measured using an accurate balance and the full 1 g of stool was extracted. For preparation of these samples, a smaller volume of sodium phosphate buffer (528 μL) was used to reserve room in the tube for the additional stool. These samples were homogenized for 80 seconds using the FastPrep Instrument on a speed setting of 6 and were then subjected to DNA extraction as described above.

### Comparison of index and previously published real-time PCR assay performance

To assess comparative performance, the sensitivity of our real-time PCR index assay was compared to that of previously published reference assays [17–19]. Testing was done using *S. japonicum* gDNA as template, as well as DNA extracted from human stool spiked with *S. japonicum* eggs. When conducting assays targeting SjTR1, the mitochondrial NADH dehydrogenase I [17] or putative DNA photo-lyase [18] genes, testing was performed in 7 μL reaction volumes containing 3.5 μL of TaqPath ProAmp Master Mix. When targeting the mitochondrial DNA 16S rRNA gene [19], reactions were performed in 5 μL volumes containing 2.5 μL of PrimeTime Gene Expression Master Mix (Integrated DNA Technologies). Testing using all published assays utilized primer and probe concentrations and annealing temperatures described in the literature. Each assay was tested against 2 ng and 200 pg of gDNA template with five [17–18] or three [19] replicates respectively run for each assay and each template

mass. When testing against spiked stool samples, three reaction replicates were run for each assay.

### Validating the SjTR1 assay using clinical samples collected from endemic area

A panel of 100 human stool samples collected from an *S. japonicum* endemic area in the Philippines, of which 38 were positive and 62 were negative based on Kato-Katz data, was used to validate the performance of our assay on clinical samples [28]. DNA from these samples was extracted using the FastDNA Spin Kit for Soil (MP Biomedicals). To evaluate the efficiency of each extraction, an internal control IAC plasmid (100 pg) was spiked into each extracted sample prior to the DNA binding step [29]. The recovery of the IAC plasmid was tested via real-time PCR in duplicate 7 uL reactions with 125 nM primer/probe concentrations and an annealing temperature of 59˚C. All samples with mean Cq values that were >3 standard deviations from the mean were re-extracted and re-tested. To assess the performance of our index assay in comparison to a published real-time PCR reference assay, we tested all samples using the published mitochondrial NADH dehydrogenase I gene-targeting assay [17]. This assay was chosen for comparison due to its performance during the spiking experiments described above. For both assays, samples that showed inconsistent amplification (amplifying in one of two replicates) as well as samples that had a standard deviation >3 between tested replicates underwent re-testing, again in duplicate, and retest results were reported. Each experiment was run for 40 cycles and a sample was considered positive if it had a Cq of 40 or less.

## Results

### Target identification and real-time PCR assay design

RepeatExplorer2 and TAREAN analyses resulted in 5.95% of the analyzed *S. japonicum* sequence reads mapping to putative satellite repeats yielding a total of 40 clusters and superclusters. From among the 24 clusters classified as "high confidence" satellites, the cluster comprising the highest proportion of the analyzed sequences (1.7%) was chosen as our assay target. The consensus sequence resulting from this cluster, SjTR1, had a length of 101 base pairs and contained no protein-coding domains. The primers and probe designed for the assay had a GC content of 50% and produced an amplicon of 80 nucleotides (Table 1 and Fig 1).

### Real-time PCR assay validation and optimization

Optimization reactions resulted in an ideal annealing/extension temperature of 59˚C (S2 Table). Primer limiting reactions determined that primers performed optimally with the forward primer at a concentration of 62.5 nM and the reverse primer at a concentration of 500 nM (S3 Table). Dilutions of genomic DNA showed the assay to be reliably sensitive at template masses as low as 200 ag (Tables 2 and S4). The assay did not amplify the DNA of any of the common gut parasites, *S. mansoni*, *S. haematobium*, the Mock Microbial Community, *C. albicans*, *E. coli*, or human DNA in the specificity panel. The assay did, however, amplify *S. mekongi* DNA (S5 Table).

### Real-time PCR assay efficiency

Using the size of the *S. japonicum* repeat-containing plasmid to estimate the number of target copies in successive 10-fold dilutions of the plasmid, a linear curve was constructed as described previously [26]. The slope of the line (-3.26) was used to calculate an assay efficiency of 102.7% with an amplification factor of 2. The $R^2$ value of the linear curve was 0.983 (Fig 2).

**Table 2. Analytical limits of detection for the SjTR1 assay.**

| Mass of *S. japonicum* DNA | Mean Cq +/- SD |
|---|---|
| 200 pg | 11.79 +/- 0.11 |
| 20 pg | 14.93 +/- 0.08 |
| 2 pg | 17.79 +/- 0.36 |
| 200 fg | 21.52 +/- 0.21 |
| 20 fg | 25.21+/- 0.62 |
| 2 fg | 28.71 +/- 0.18 |
| 200 ag | 32.14 +/-1.47 |
| 20 ag | N/A |
| 2 ag | N/A |

## Comparison of *S. japonicum* real-time PCR assay targets

Real-time PCR comparing the detection of *S. japonicum* assay targets using *S. japonicum* gDNA showed that the mean Cq values produced by the SjTR1-targeting assay were 16.45, 20.66, and 14.21 cycles lower than those produced by previously published assay targets when testing 2 ng of template DNA and 15.10, 19.92, and 19.48 cycles lower when testing 200 pg of template DNA (Fig 3 and S7 Table) [17–19]. Although comparisons made using genomic DNA as template cannot be used to evaluate the comparative clinical sensitivities of the examined assays, these sizable differences in mean Cq values strongly suggest that the SjTR1 is considerably more abundant in copy number within the *S. japonicum* genome than previously published targets [17–19,21]. When testing stool spiked with *S. japonicum* eggs, the mean Cq values produced by the SjTR1-targeting assay were 12.70, 17.86, and 12.09 cycles lower than values produced by previously published assays on samples containing 20 epg of stool; 12.78, 17.77, and 12.27 cycles lower on samples containing 60 epg of stool; and 12.24, 17.74, and 11.75 cycles lower on samples containing 200 epg of stool [17–19] (Tables 3 and S8). Most notably, when testing 1 gram stool samples spiked with a single egg, the SjTR1-targeting assay

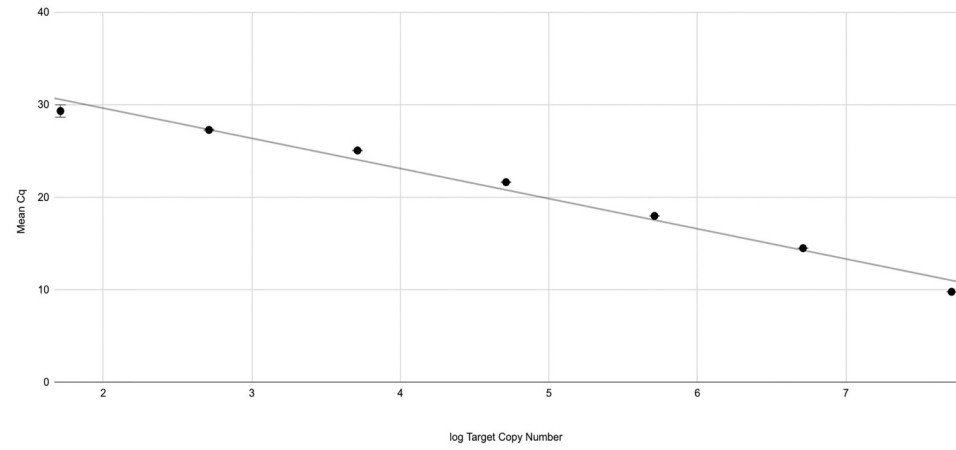

**Fig 2. Assay efficiency.** Ten-fold serial dilutions of the plasmid, ranging from 100 pg to 100 ag were prepared and the target copy number was estimated for each dilution (S6 Table). The log of the target copy number was plotted against the mean Cq of 11 replicates for each respective dilution. The slope of the line was used to determine the efficiency of the assay. Error bars are included but due to the small standard deviations resulting from each concentration of plasmid DNA tested, they are not distinguishable at most points.

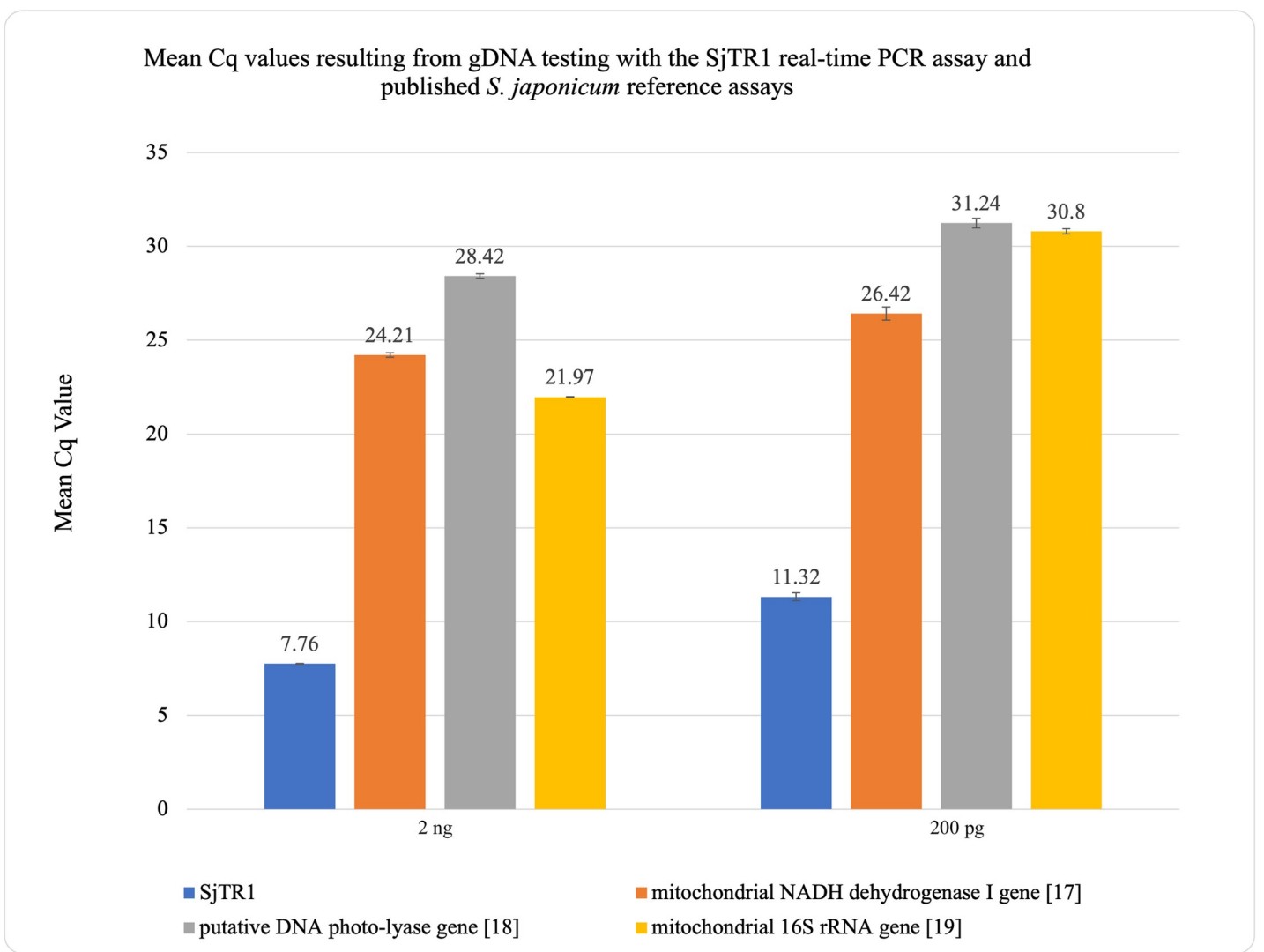

**Fig 3. Comparison of mean Cq values.** Results from gDNA testing with the SjTR1 real-time PCR assay and previously published *S. japonicum* assays.

detected *S. japonicum* DNA in all tested samples, while the mitochondrial NADH dehydrogenase I gene target assay [17] detected DNA in only eight of the ten samples, with sporadic detection in four of these replicates. All other assays failed to detect DNA from any of the "one egg" samples (Tables 4 and S9). Ten negative controls were analyzed in parallel, containing 1 gram of naive stool and no *S. japonicum* eggs. No negative control samples were amplified by any of the assays.

### Validation of the assay on clinical samples

A panel of 100 field collected samples from endemic areas in the Philippines were blindly tested, of which 38 were positive and 62 were negative based on Kato-Katz data [28]. Testing of our SjTR1 real-time PCR assay resulted in 59 positive samples and 41 negative samples (S10 Table). On the other hand, testing using the NADH dehydrogenase I real-time PCR assay [17] resulted in 52 positive samples and 48 negative samples (S10 Table). Seven of the SjTR1 positive samples were not detected by the NADH dehydrogenase I assay while all 41 negatives by

**Table 3. Comparative detection of DNA extracted from naive human stool spiked with *S. japonicum* eggs.**

|  | 1 egg[a] = 20 EPG Mean Cq [Range] | 3 eggs[a] = 60 EPG Mean Cq [Range] | 10 eggs[a] = 200 EPG Mean Cq [Range] |
|---|---|---|---|
| SjTR1 | 16.68 [15.93–17.19] | 15.58 [14.89–15.99] | 14.06 [13.60–14.34] |
| mitochondrial NADH dehydrogenase I gene [17] | 29.38 [28.94–30.00] | 28.36 [26.73–29.46] | 26.30 [25.71–26.73] |
| putative DNA photo-lyase gene [18] | 34.54 [33.38–35.96] | 33.35 [32.36–33.88] | 31.80 [31.31–32.25]] |
| mitochondrial 16S rRNA gene [19] | 28.77 [28.44–29.17] | 27.85 [26.42–28.95] | 25.81[25.32–26.19] |

[a] Number of eggs spiked in a sample containing 0.05 g of stool.

Each concentration of eggs was tested using three biological replicates, and each replicate was analyzed by the SjTR1 assay in triplicate. The reported mean Cq value was calculated as a mean value of all component sample means. The reported range includes the smallest and greatest individual Cq values for each egg concentration. EPG = eggs per gram of stool.

the SjTR1 assay were not detected by the NADH dehydrogenase I assay [17] (S10 Table). The same thirty-six of the thirty-eight Kato-Katz positive samples were amplified by both the SjTR1 and the NADH Dehydrogenase assays. The two unidentified Kato-Katz positive samples were not detected by any of the published real-time PCR assays discussed in this manuscript [17–19]. For the positive samples, the minimum, the median, the maximum, and the quartiles of mean Cq values are shown below for both assays (Fig 4). Among the positive samples, the mean Cq difference between the two assays was 10 cycles lower for the SjTR1 assay compared to the NADH dehydrogenase I assay (S10 Table).

## Discussion

Endemic in 3 provinces in Indonesia, 28 provinces in the Philippines, in Taiwan (zoophilic strain), and in China, where its infection levels exceed 10% in high-risk populations, the diagnosis of *S. japonicum* has typically occurred using the Kato-Katz technique, a method which has been shown to underestimate infection levels by up to 70% [3,9,30–32]. Immunological tests, like antigen detection tests and enzyme-linked immunosorbent assays (ELISA), are another possible avenue for diagnosis of *S. japonicum*. These tests, however, often lack specificity and sensitivity [33–34]. Such detection methods may lead to insufficient intervention efforts, making the development of highly sensitive, specific molecular diagnostic tools imperative for the successful elimination of *S. japonicum*.

Within target areas, preventive chemotherapy efforts through praziquantel mass drug administration as well as eligibility of different age groups for treatment is determined based on prevalence of infection as assessed via positive parasitological diagnosis [35]. Thus, for the Western Pacific Region and South-East Asia, the WHO regional priorities for the 2012–2020 period included maintenance of high and/or regular coverage of preventive chemotherapy in the Philippines and Lao People's Democratic Republic, intensification of preventive chemotherapy in Cambodia, China and Indonesia, and verification of status of elimination in Japan, Malaysia, Thailand, and India [35]. Of these areas, *S. japonicum*, whose elimination from its endemic areas is complicated by the presence of animal reservoir hosts, is endemic in the Philippines and China—the two countries with the highest proportion of people requiring treatment for schistosomiasis in the Western Pacific Region—and in Indonesia [9,35]. Furthermore, to determine the success of these treatment efforts, sensitive and specific diagnostic tools are needed [9,36–37].

The results reported here demonstrate that our repeat-targeting assay can reliably detect *S. japonicum* at concentrations as low as 1 egg per gram of human stool. Putting these results in

**Table 4. Comparative Detection of 1 gram stool samples spiked with an individual *S. japonicum* egg (positive samples are bolded; negative controls are in standard font).**

| | SjTR1[*] | | mitochondrial NADH dehydrogenase I gene [17][*] | | putative DNA photo-lyase gene [18][*] | | mitochondrial 16S rRNA gene [19][*] | |
|---|---|---|---|---|---|---|---|---|
| | Mean Cq [Range][a] | Total Detected | Mean Cq [Range][a] | Total Detected | Mean Cq [Range][a] | Total Detected | Mean Cq [Range][a] | Total Detected |
| **Sample 1** | **38.45 [36.96–39.95]** | **2/3** | **35.44** | **1/3** | **N/A** | **0/3** | **N/A** | **0/3** |
| **Sample 2** | **31.62 [31.42–31.95]** | **3/3** | **36.58** | **1/3** | **N/A** | **0/3** | **N/A** | **0/3** |
| Sample 3 | N/A | 0/3 | N/A | 0/3 | N/A | 0/3 | N/A | 0/3 |
| **Sample 4** | **30.05 [29.93–30.25]** | **3/3** | **37.67 [35.02–39.31]** | **3/3** | **N/A** | **0/3** | **N/A** | **0/3** |
| Sample 5 | N/A | 0/3 | N/A | 0/3 | N/A | 0/3 | N/A | 0/3 |
| Sample 6 | N/A | 0/3 | N/A | 0/3 | N/A | 0/3 | N/A | 0/3 |
| **Sample 7** | **30.90 [30.55–31.19]** | **3/3** | **36.57 [36.27–36.86]** | **2/3** | **N/A** | **0/3** | **N/A** | **0/3** |
| Sample 8 | N/A | 0/3 | N/A | 0/3 | N/A | 0/3 | N/A | 0/3 |
| Sample 9 | N/A | 0/3 | N/A | 0/3 | N/A | 0/3 | N/A | 0/3 |
| **Sample 10** | **33.45 [33.26–33.72]** | **3/3** | **N/A** | **0/3** | **N/A** | **0/3** | **N/A** | **0/3** |
| **Sample 11** | **31.93 [31.49–32.27]** | **3/3** | **35.00 [33.42–36.20]** | **3/3** | **N/A** | **0/3** | **N/A** | **0/3** |
| **Sample 12** | **30.34 [30.08–30.49]** | **3/3** | **33.97 [33.58–34.40]** | **3/3** | **N/A** | **0/3** | **N/A** | **0/3** |
| Sample 13 | N/A | 0/3 | N/A | 0/3 | N/A | 0/3 | N/A | 0/3 |
| **Sample 14** | **33.08 [32.87–33.45]** | **3/3** | **N/A** | **0/3** | **N/A** | **0/3** | **N/A** | **0/3** |
| **Sample 15** | **31.06 [30.85–31.34]** | **3/3** | **35.85** | **1/3** | **N/A** | **0/3** | **N/A** | **0/3** |
| Sample 16 | N/A | 0/3 | N/A | 0/3 | N/A | 0/3 | N/A | 0/3 |
| Sample 17 | N/A | 0/3 | N/A | 0/3 | N/A | 0/3 | N/A | 0/3 |
| Sample 18 | N/A | 0/3 | N/A | 0/3 | N/A | 0/3 | N/A | 0/3 |
| Sample 19 | N/A | 0/3 | N/A | 0/3 | N/A | 0/3 | N/A | 0/3 |
| **Sample 20** | **33.15 [32.81–33.34]** | **3/3** | **37.16** | **1/3** | **N/A** | **0/3** | **N/A** | **0/3** |

a. Each spiked sample was run in triplicate, and the mean Cq values are reported. The reported range includes the smallest and greatest individual Cq values for each individual sample. None of the negative control samples were amplified by any of the assays.

*The SjTR1 assay detected *S. japonicum* DNA in all 10 of the samples spiked with a single egg (amplification in three out of three replicates for 9 of the 10 samples and two out of three for one of the samples). The mitochondrial NADH dehydrogenase I real-time PCR assay [17] detected *S. japonicum* DNA in only eight of the ten samples tested, with sporadic detection (amplification of only one out of three replicates) in four of these samples. The real-time PCR assays targeting the putative DNA photo-lyase and mitochondrial 16S rRNA genes [18–19] did not detect *S. japonicum* DNA in any of the samples.

context, *S. japonicum* infection intensity is defined by the World Health Organization as light (1–100 eggs per gram), moderate (101–400 eggs per gram), and heavy (more >400 eggs per gram) [38]. As such, this assay should greatly improve detection capabilities in areas of low infection intensity if used for determination of infection prevalence.

For the accurate estimation of parasite prevalence necessary to shape and guide mass treatment efforts, the specificity of a diagnostic method is also critical. Accordingly, one of our assay's main limitations is its ability to detect the closely related Asian schistosome, *S. mekongi*. However, the range of *S. mekongi* is believed to be limited to select areas along the Mekong River Basin within the Lao People's Democratic Republic and Cambodia [39–40]. In addition, *S. japonicum* and *S. mekongi* are only thought to overlap in a limited region in Myanmar [41]. Thus, in this area of possible co-endemicity, a positive test result from our highly sensitive assay would need to be confirmed using methods allowing for the molecular differentiation of *S. mekongi* and *S. japonicum* [42–43]. It is worth noting that our assay can reliably detect as little as 2 fg of *S. mekongi* gDNA, which provides evidence for the possible use of the SjTR1 assay for the detection of *S. mekongi*. The suitability of our assay for detection of *S. mekongi* in human stool as well as optimization of real-time PCR for differential detection of the two species will be explored in a future study. Further work will also test the SjTR1 assay against various *S. japonicum* and *S. mekongi* strains found in Asia. Although mass drug administration for all blood flukes causing schistosomiasis relies on the same drug, praziquantel, specificity of

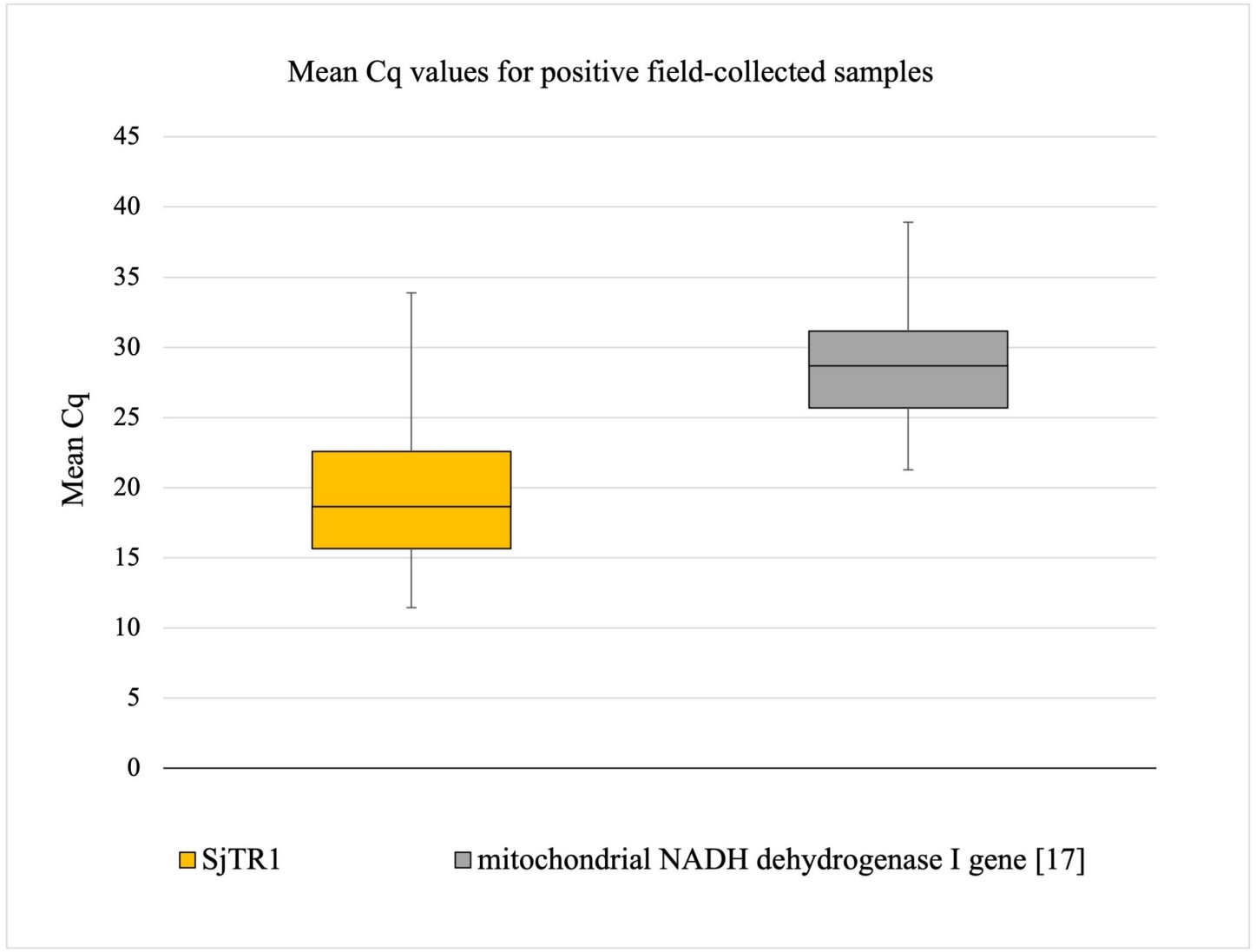

**Fig 4. Boxplots for mean Cq values of positive field collected samples.** Results for testing with the SjTR1 and mitochondrial NADH dehydrogenase I gene [17] assays.

testing is nonetheless important for accurate surveillance of parasite endemicity and infection prevalence.

When compared to other published real-time PCR assays, testing of the SjTR1 real-time PCR assay described here demonstrated remarkable reductions in Cq values and therefore increased sensitivity. This suggests that the target of the assay, a putative tandem repeat, is more abundant in the *S. japonicum* genome than the target regions of the other qPCR assays tested. This provides additional support for improved analytical sensitivity via the use of satellite repeats instead of traditional targets such as ribosomal or mitochondrial DNA targets [20–21]. Evidence of the increased clinical sensitivity for our assay compared to previously published assays is supported by the results of the spiking study in which the SjTR1 assay was the only real-time PCR assay capable of reliably detecting 1 egg per gram in all samples (Table 4).

Providing further evidence of increased clinical sensitivity, our assay also showed superior performance when used to test field-collected clinical stool samples from an endemic area in the Philippines. Outperforming the most promising published real-time PCR assay through

the detection of seven additional samples, our assay detected twenty-one samples more than the traditional Kato-Katz technique. We were not surprised to see that two Kato-Katz positive samples (containing 35 and 113.3 epg, respectively) were not detected by any of the four real-time PCR assays using various targets discussed in the study (S10 Table). Given the known specificity challenges associated with the Kato-Katz technique and the lack of amplification by four sensitive real-time PCR assays, we strongly suspect that those two samples represent Kato-Katz false positive results. Our assay was able to detect parasite DNA in clinical samples with as few as 3 epg, providing evidence that sensitivity of our assay seems like an unlikely cause for these results. Furthermore, the internal control results give us confidence that the quality of the extraction did not interfere with the results. Nonetheless, these two samples comprise an important avenue for further exploration and will be examined using next generation DNA sequencing in a future study. For the positive samples, the mean Cq difference between our assay and the NADH dehydrogenase assay [17] was 10 cycles, providing additional evidence for our assay's superior sensitivity. Although these results provide strong evidence for the clinical utility of our assay, given *S. japonicum*'s wide range of hosts, further validation of the assay on stool samples from different endemic areas as well as multiple animal hosts will be important.

Given the acknowledged shortcomings of current coproscopic and immunological methods employed for schistosome detection, improved diagnostic methods for *S. japonicum* are urgently needed. Due to increased need for HIV, and more recently COVID-19, real-time PCR testing, the availability of updated PCR testing labs in endemic countries has greatly increased, facilitating the use of cost-effective, real-time PCR assays for detection of numerous infectious agents. In fact, it has been shown that the cost of PCR is comparable to that of Kato-Katz with both double-slide Kato-Katz and duplicate PCR testing of a sample costing around 2 US dollars [44–45]. In conclusion, this assay has the capacity to improve detection of *S. japonicum*, helping to guide programmatic decision-making efforts to control and eventually eliminate *S. japonicum* from endemic countries.

## Supporting information

**S1 STARD Checklist. Checklist indicating where criteria for assessing potential study biases have been addressed.**
(DOCX)

**S1 Table. Supplier/origins of parasites tested in specificity panel.**
(XLSX)

**S2 Table. Cq values from annealing temperature optimization experiment.**
(XLSX)

**S3 Table. The results of primer optimization reactions for SjTR1 assay.** The Cq values are provided for each replicate of forward and reverse primers diluted at concentrations ranging from 62.5 nM to 1000 nM in all possible combinations.
(XLSX)

**S4 Table. Cq values for each replicate for analytical limits of detection for the SjTR1 assay on *S. japonicum* gDNA, ranging from 200 pg to 2 ag.**
(XLSX)

**S5 Table. Cq values for each replicate for analytical limits of detection for the SjTR1 assay on *S. mekongi* gDNA, ranging from 200 pg to 20 ag.**
(XLSX)

**S6 Table. Cq values for each replicate of plasmid target copy number used to calculate the assay's efficiency.**
(XLSX)

**S7 Table. Cq values for each replicate resulting from gDNA testing with the SjTR1 real-time PCR assay and published *S. japonicum* assays.**
(XLSX)

**S8 Table. Cq values for each replicate of comparative detection of DNA extracted from naive human stool spiked with *S. japonicum* eggs using SjTR1 and published assays.**
(XLSX)

**S9 Table. Cq values for each replicate of each sample from comparative detection of 1 gram stool samples spiked with an individual *S. japonicum* egg using SjTR1 and published assays.**
(XLSX)

**S10 Table. Cq values for each replicate for experimental data from testing SjTR1 and NADH dehydrogenase I assays on clinical stool samples, Kato-Katz data, and IAC control data.**
(XLSX)

## Acknowledgments

We would like to thank Ms. Kareen Seignon, Susan Haynes, Dr. Samantha D. Torquato, and Dr. Lori Saunders for their assistance. Computational resources were provided by the CERIT-SC Center (LM2015085) and ELIXIR-CZ project (LM2015047), part of the international ELIXIR infrastructure. Genomic DNA from the Microbial Mock Community B (Staggered, High Concentration), v5.2H, for Whole Genome Shotgun Sequencing, HM-277D was obtained through BEI Resources, NIAID, NIH as part of the Human Microbiome Project. *S. japonicum* eggs were provided by the NIAID Schistosomiasis Resource Center of the Biomedical Research Institute (Rockville, MD) through NIH-NIAID contract HHSN2722010000051. We'd also like to thank Dr. Aidan Emery, Dr. Fiona Allan, and Ms. Muriel Rabone at SCAN, Schistosomiasis Collection at the Natural History Museum, which is part-funded by the Wellcome Trust (grant 104958/Z/14/Z), for providing us with *S. japonicum*, *S. mansoni*, and *S. haematobium* samples. Material provided via SCAN was originally collected between 1993 and 2015 and we acknowledge the support and generosity of our partners from the countries concerned. We would like to thank Dr. Poom Adisakwattana at Mahidol University, Salaya, Thailand for graciously providing us with *S. mekongi* worms and Dr. Thomas Nutman at the National Institutes of Health (Bethesda, MD) for providing us with *T. solium* and *T. crassiceps*.

## Author Contributions

**Conceptualization:** Sara Halili, Jessica R. Grant, Nils Pilotte, Steven A. Williams.

**Funding acquisition:** Steven A. Williams.

**Investigation:** Sara Halili, Jessica R. Grant.

**Methodology:** Sara Halili, Jessica R. Grant, Nils Pilotte, Steven A. Williams.

**Resources:** Catherine A. Gordon.

**Writing – original draft:** Sara Halili.

**Writing – review & editing:** Sara Halili, Jessica R. Grant, Nils Pilotte, Catherine A. Gordon, Steven A. Williams.

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
