## [Decision Letter · Decision Letter 0]

13 Apr 2021

Dear Ms. Halili,

Thank you very much for submitting your manuscript "Development of a novel quantitative polymerase chain reaction assay for the sensitive and species-specific detection of Schistosoma japonicum in human stool" for consideration at PLOS Neglected Tropical Diseases. As with all papers reviewed by the journal, your manuscript was reviewed by members of the editorial board and by several independent reviewers. In light of the reviews (below this email), we would like to invite the resubmission of a significantly-revised version that takes into account the reviewers' comments. 

We cannot make any decision about publication until we have seen the revised manuscript and your response to the reviewers' comments. Your revised manuscript is also likely to be sent to reviewers for further evaluation.

Sincerely,

Cinzia Cantacessi

Deputy Editor

Makedonka Mitreva

Deputy Editor

Reviewer's Responses to Questions

**Key Review Criteria Required for Acceptance?**

**Methods**

-Are the objectives of the study clearly articulated with a clear testable hypothesis stated?

-Is the study design appropriate to address the stated objectives?

-Is the population clearly described and appropriate for the hypothesis being tested?

-Is the sample size sufficient to ensure adequate power to address the hypothesis being tested?

-Were correct statistical analysis used to support conclusions?

-Are there concerns about ethical or regulatory requirements being met?

Reviewer #1: The paper focuses on the analytical design and testing of a qPCR for S. japonicum. For Plos NTD the methods need more detail. Point details are shown below but a careful read and editing by the authors would help bring clarity to the methods and to provide the detail needed by the readers. 

• Page 7 – first paragraph. For the samples from the NHM – were these from worms or miracidia? If from worms why was WGA done? Also the geographical origin would be useful. 

• Validation and Optimization – last sentence – which, genomic DNA was used. ? S. japonicum. 

• Specificity section page 7 and 8 – the origin / supplier of the samples / DNA tested needs to be added. 

• Plasmid control – add some more detail on how the target sequence was generated. PCR and then clean up etc.? How many repeats were in the target cloned. Also, a little more detail to this. How the colonies were selected, screened and purified etc. 

• Explain more on how you go from a plasmid to the working concentration. This will not be common knowledge for many readers. 

• Testing naïve stool samples – where were the eggs from, state that they are S. japonicum and say how you counted / selected them. How was the weight of the stool measured and also I could not understand the addition of buffer to reserve room in the tube and was the whole 1g sample extracted? 

• Figure 1 is confusing. You have highlighted the forward primer and the probe on the reverse strand. Although it is not incorrect it is confusing as I am not sure why you would not just highlight them on the forward strand and the reverse primer on the reverse strand? Also the two strands are not aligned so it does not make it easy to look at. Just have one strand in the forward direction and then mark the primers and probe and the sequence.

Reviewer #2: I feel that the number of species of gut microbes used in cross-reaction testing is insufficient. At a bare minimum, this must be tested against Schistosoma mekongi DNA. I understand that the authors have had difficulty obtaining this, but I encourage them to redouble their efforts. Researchers have published the transcriptome, are there at least in silico sequences that could be tested? I see this as a major deficiency in the paper.

Further cross-reaction testing against Giardia duodenalis, a series of other Enterobacteriaceae, Enterococcus spp., Taenia spp., Necator americanus, Clonorchis sinensis and Opisthorchis viverrini, should be performed. DNA of the former five species/groups are easily obtained, so I am surprised that this has not already been performed.

As this assay may well be used on animals by other researchers, if possible, it should be tested on animal schistosomes as well, particularly S. malayensis of rats and S. incognitum of rats, dogs and pigs, both of which may cause rare human zoonotic infections in Asia.

Where was the NHM S. japonicum DNA used from geographically? S. japonicum from Indonesia, China and the Philippines are quite genetically distinct. The Formosan (Taiwanese) strain of S. japonicum, which only infect animals, are genetically distant from other S. japnonicum. You must state the origin of your S. japonicum DNA and eggs used in this study.

Reviewer #3: This manuscript describes a newly designed species-specific real-time PCR for the detection of Schistosoma japonicum DNA in human stool. The authors claim that by cleverly designing the right target (in this case: Schistosoma japonicum Tandem Repeat 1), their PCR is more sensitive than previously published real-time PCR assays for the detection of S. japonicum.

I found the manuscript interesting to read and the PCR described seems to have great potential. My major criticism is that the authors did not use the opportunity to test their PCR on a collection of field samples. In my opinion this would have greatly benefitted the impact of the publication. I find it hard to imagine these authors have no access at all to a set of clinical stool samples, even if only a small one, to prove the diagnostic value of their PCR. A set of samples collected before and after treatment with praziquantel would have been ideal to illustrate the clinical value of their PCR.

Some minor points concerning methods:

Page 7: ” DNA was extracted from samples of S. japonicum, S. mansoni, and S. haematobium graciously provided by the Natural History Museum, London, UK” – more details about the original samples would have been informative.

Page 8 bottom: I would like to read more details of the source of the S. japonicum eggs. Do these eggs fully represent the eggs normally seen in freshly collected stool samples?

**Results**

-Does the analysis presented match the analysis plan?

-Are the results clearly and completely presented?

-Are the figures (Tables, Images) of sufficient quality for clarity?

Reviewer #1: The results need some detail added and also need a careful read to make sure nothing is missing. Also as required by Plos NTD all qC values should be provided for all replicates and all samples, even the controls. 

• Describe what happened at temperatures above and below 59. 

• Was there a cut off Cq score for non-amplification used to test the LOD and the specificity? You do not mention S. mansoni or S. haematobium in the specificity section. 

• It would be better to refer to the published targets that you tested as published targets rather than reference targets and make it clear that these are published and not designed as part of this study. 

• You mention the negative controls in the results but all the negative controls and how they were set up need to be described in the methods. Negative controls also need to be described for the validation steps.

Reviewer #2: No specific comments

Reviewer #3: See above.

**Conclusions**

-Are the conclusions supported by the data presented?

-Are the limitations of analysis clearly described?

-Do the authors discuss how these data can be helpful to advance our understanding of the topic under study?

-Is public health relevance addressed?

Reviewer #1: Although the discussion supports the data and the study it is very brief and does not address the wider context of the study. Suggestions for improvement are shown below. 

Line three change “historically occurred” to “typically performed” as you should refer to what is typically done now. 

• It would be good to provide some information on the interventions in S. japonicum areas. Is routine mass testing done or is it done on a case by case basis, is MDA performed or is it a test and treat scenario. The wider context for the use of qPCR as a test needs to be discussed particularly in relation to other tests such as serology and antigen. What would be the application of a qPCR test in these settings? 

• Page 15 – when talking about the critical need for specificity you could include the recent Plod NTD paper by Katie Gass. 

• When talking about the need to look at field samples you should also mention other variables such as would the presence of CFPD have an impact? 

• You should also talk about the endemic setting and how such a qPCR could fit. For example in S. japonicum settings are labs available for the testing or are portable systems needed like in Africa and also some discussion around cost is needed. Does the test need to fit TPP’s of schisto? 

• You should also talk about the diagnostic in terms of intensity analysis – limitations or if it can be used as a quantitative assay or not? 

• Also, talk about the potential utility in animal and snail hosts.

Reviewer #2: The greatest issue with the discussion as it stands is the attempt to minimise the need for further validation of this assay in an endemic setting on both human and animal samples. This must be amended and the deficiencies of not having done this very clearly and openly acknowledged and explored.

Discussion first sentence: Also endemic in Taiwan (zoophilic Formosan strain)

If the authors do not test against the many genotypes of S. japonicum found in Asia, they must clearly state that their assay described here has not been validated against the multiple genotypes possible and that this work should be performed in the future.

Similarly, if the authors do not test against the S. japonicum from multiple animal hosts and multiple human hosts, they have not entirely validated their assay and they must state such testing is required to validate the assay before any clinical, public health and particularly veterinary use.

Page 16 first para. I do not believe that the spiking of S. japonicum eggs into naive human stool samples provides a close approximation of real samples since our mock samples. The point of such validations is to test multiple isolates from different hosts as well as to approximate its validity in the faecal matrix. Please significantly amend this sentence to reflect my comment here. 

Page 16 para 1: Overall, this assay needs to be compared to Kato Katz and some of the other PCRs against a statistically valid number of human and animal samples in an endemic setting, preferably three endemic settings (one in Indonesia, one in China and one in Indonesia). I understand that this work may yet be performed, but I think it is important that the authors better acknowledge the need for such validation very clearly in the discussion, please amend the sentence addressing the need for field validation to better.

Line 15: I see the failure to test the assay against S. mekongi as a very significant limitation. You should remove the sentence saying that it is not and acknowledge that validation is not complete without such testing.

Reviewer #3: As mentioned above, I find the overall content of this manuscript rather limited. As the authors have mentioned, the focus is entirely on the technical characteristics of the PCR, without any clinical validation. This is a serious omission. One of the major advantages of S. japonicum is the fact that this parasite also affects various non-human hosts, such a cattle. So even if the authors have no access to human stool, they could have tested naturally infected animal samples.

Another limitation of the manuscript is that the authors suggest throughout the manuscript that PCR is the only diagnostic alternative to microscopic detection of S. japonicum eggs. In fact, detection of circulating antigen in serum or urine, a more field-friendly procedure for diagnosing an active S. japonicum infection, has also shown to be a promising alternative. See: Van Dam et al., (2015) Evaluation of banked urine samples for the detection of circulating anodic and cathodic antigens in Schistosoma mekongi and S. japonicum infections: a proof-of-concept study. Acta Trop. 2015 Jan;141(Pt B):198-203.

**Editorial and Data Presentation Modifications?**

Reviewer #1: A careful read and careful revision of terminology would help improve the paper. Some sections are hard to follow and could be written better and with more detail to help interpretation.

Reviewer #2: The results section is very "table heavy" – this could be remedied by changing table 3 into a figure.

page 12 first para - no space between genus and species name for "S. japonicum"

Page 15 final para - italicise S. mekongi

Reviewer #3: See above.

**Summary and General Comments**

Reviewer #1: This is a very informative paper that describes the development and lab. testing of a S. japonicum qPCR assay to support diagnosis. This will be a very useful assay due to it superior sensitivity. The way the target was identified and used shows substantial progress in the design of molecular assays and the use of genomic data that is now becoming available. This will also support the design of assays for other species. The paper needs more detail and a more wider discussion to bring it up to the standard required for Plos NTD.

Some points on the author summary and intro are shown below.

Please add line number as this helps with reviewing. 

• Author Summary – 230 million is for schisto total not for S. japonicum – revise the first sentence. This should also be made clear in the first line of the introduction. 

• Page 4 last paragraph – add stool or faecal before samples. Line 5 state if this is egg derived DNA or CFPD or both.

Reviewer #2: This is a worthwhile assay but further validation against other common gut parasites found in S. japonicum endemic regions is needed.

Furthermore, if S. mekongi DNA cannot be obtained, or sequences analysed in silico, this must be acknowledged as a significant deficiency.

The authors tend to use terms such as "we do not feel this is a significant limitation" when describing deficiencies of their own work. This not only gets a reviewer's "back up" as it is not for the authors to decide what is an is not a significant limitation of their approach, it is also not good scientific writing. I suggest simply acknowledging your work's deficiencies without attempting to editorialize on how important or unimportant they are.

Reviewer #3: This manuscript is well written, the objectives are clear and the PCR described is potentially relevant for those working in the field of S. japonicum. Never the less, the study would benefit from some additional experimental work, in particular testing of real clinical samples.

PLOS authors have the option to publish the peer review history of their article (what does this mean?). If published, this will include your full peer review and any attached files.

Reviewer #1: No

Reviewer #2: No

Reviewer #3: No
---

## [Decision Letter · Decision Letter 1]

3 Oct 2021

Dear Ms. Halili,

Thank you very much for submitting your manuscript "Development of a novel real-time polymerase chain reaction assay for the sensitive detection of Schistosoma japonicum in human stool" for consideration at PLOS Neglected Tropical Diseases. As with all papers reviewed by the journal, your manuscript was reviewed by members of the editorial board and by several independent reviewers. The reviewers appreciated the attention to an important topic. Based on the reviews, we are likely to accept this manuscript for publication, providing that you modify the manuscript according to the review recommendations. 

Sincerely,

Cinzia Cantacessi

Deputy Editor

Makedonka Mitreva

Deputy Editor

Reviewer's Responses to Questions

**Key Review Criteria Required for Acceptance?**

**Methods**

-Are the objectives of the study clearly articulated with a clear testable hypothesis stated?

-Is the study design appropriate to address the stated objectives?

-Is the population clearly described and appropriate for the hypothesis being tested?

-Is the sample size sufficient to ensure adequate power to address the hypothesis being tested?

-Were correct statistical analysis used to support conclusions?

-Are there concerns about ethical or regulatory requirements being met?

Reviewer #2: (No Response)

**Results**

-Does the analysis presented match the analysis plan?

-Are the results clearly and completely presented?

-Are the figures (Tables, Images) of sufficient quality for clarity?

Reviewer #2: (No Response)

**Conclusions**

-Are the conclusions supported by the data presented?

-Are the limitations of analysis clearly described?

-Do the authors discuss how these data can be helpful to advance our understanding of the topic under study?

-Is public health relevance addressed?

Reviewer #2: (No Response)

**Editorial and Data Presentation Modifications?**

Reviewer #2: (No Response)

**Summary and General Comments**

Reviewer #2: The authors have now tested their assay against S. mekongi, which was my major concern upon review of the original manuscript. The have addressed my other comments well. 

While the reviewers have now included the zoophilic S. japonicum from Taiwan, it has not been acknowledged as only infecting animals. I would like to suggest the authors include a sentence noting this. I think it is important, as otherwise an error may enter the literature with this article being cited by the uninformed as describing human infection from/in Taiwan.

PLOS authors have the option to publish the peer review history of their article (what does this mean?). If published, this will include your full peer review and any attached files.

Reviewer #2: No

Figure Files:

Data Requirements:

Reproducibility:

References

---

## [Editor Report · Decision Letter 2]

6 Oct 2021

Dear Ms. Halili,

We are pleased to inform you that your manuscript 'Development of a novel real-time polymerase chain reaction assay for the sensitive detection of Schistosoma japonicum in human stool' has been provisionally accepted for publication in PLOS Neglected Tropical Diseases.

Best regards,

Cinzia Cantacessi

Deputy Editor

Makedonka Mitreva

Deputy Editor

---

## [Editor Report · Acceptance letter]

21 Oct 2021

Dear Ms. Halili,

We are delighted to inform you that your manuscript, "Development of a novel real-time polymerase chain reaction assay for the sensitive detection of Schistosoma japonicum in human stool," has been formally accepted for publication in PLOS Neglected Tropical Diseases.

Best regards,

Shaden Kamhawi

co-Editor-in-Chief

Paul Brindley

co-Editor-in-Chief
